# Using Land to Promote Refugee Self-Reliance in Uganda

Timothy Berke *  and Larissa Larsen

Taubman College of Architecture and Urban Planning, University of Michigan, 2000 Bonisteel Boulevard, Ann Arbor, MI 48109-2069, USA; larissal@umich.edu
* Correspondence: tberke@umich.edu

**Abstract:** Around the world, the number of people displaced from their homes continues to increase due to political conflict and climate change. The Ugandan government's policy for settling refugees shifts the focus from seeing refugees as humanitarian obligations to recognizing that refugee settlements bring improved services and infrastructure that can serve both refugees and residents of the host communities. A key aspect of this policy is to provide refugees with land use rights, so they are able to achieve "self-reliance". This research investigates (1) the role of planning in establishing the physical layout of Bidi Bidi, Uganda's largest integrated settlement, (2) the host community's rationale for allowing refugees access to their land, and (3) how refugees used this land. Based on interviews, participant observation, GIS analysis, and document analysis, we learned how refugees altered the settlement's layout to better meet their needs, the unintended consequences of large-scale deforestation on the most vulnerable, and the limitations of the land-based approach to achieve self-reliance. These findings suggest expanding the avenues to achieve self-reliance, improving "integration" of service provision between refugees and residents of the host community, and protecting the area's environment to maintain needed resources.

**Keywords:** refugee; land use; integration; self-reliance; host community; Uganda

## 1. Introduction

Globally, the number of persons displaced by persecution, violence, technological disasters, or weather-related disasters is rising. Currently, 79.5 million people are displaced from their homes with a subset of 33.8 million people also displaced from their country [1].

According to the United Nations' High Commission for Refugees (UNHCR), the UN's Refugee Agency, approximately forty percent of displaced persons live in refugee camps/settlements [2]. While refugee camps efficiently offer short-term emergency aid, many residents remain for extended periods. From 1993 to 2003, the average duration of major refugee situations has increased from 9 to 17 years [3]. The majority of refugee camps are located near host communities whose residents resemble the refugees in their extreme poverty, lack of basic infrastructure and social programs, and frequent experiences of food scarcity. Unruh writes, "The enormity of the African refugee problem underscores the importance of resettlement issues in land use planning" [4] (p. 49).

Uganda has the largest number of displaced persons in Africa and the third-largest number in the world [5]. More than 1.4 million refugees or asylum seekers live in Uganda and most come from South Sudan, Somalia, Burundi, and the Democratic Republic of Congo (DRC) [5]. Generally, in a traditional camp, refugees are denied freedom of movement and legal employment, and live in segregated areas away from surrounding communities. Starting more than 20 years before, Ugandan officials formalized their practice of creating "integrated settlements" in 2006 [6]. The integrated settlement approach recasts the notion of a traditional camp from a humanitarian obligation that consumes resources [7] to a component of regional development that attracts investment. Unlike traditional camps, integrated settlements are located adjacent to a host community's settlement and offer residents of the host community access to the camp's services, schools, and infrastructure

upgrades [8]. Refugees living in integrated settlements are permitted freedom of movement, legal employment, access to the host community's markets, and use rights to allocated plots of land. Integrated settlements are intended to foster refugee self-reliance.

The existing literature on Uganda's refugee policy, integrated settlements, and refugees' achievement of self-reliance (from before and after the 2006 Refugee Act) fails to specifically address land negotiations and acquisition, pre-construction planning efforts, and the settlement's physical layout [6,9–16]. Bidi Bidi is the largest integrated settlement in Northern Uganda. We use this as an exemplary case because of its scale and its recent 2016 establishment. This research is based upon 13 interviews with tribal leaders, local and national planners, public officials, and employees of aid organizations conducted in 2019, five site visits from 2016–2019, participant observation in the settlement and with service provision coordination meetings, and a review of relevant documents (e.g., UNDP evaluation reports and humanitarian and governmental reports) and academic literature.

This article has three purposes. The first purpose is to describe the Yumbe District of Uganda and the host community before the arrival of South Sudanese refugees in 2016. Then, based on our interviews and participant observations, we explain how and why the local leaders gave the refugees use rights to their communal land. The second purpose is to describe the construction process of Bidi Bidi and analyze the challenges associated with planning and implementation. We supplement this text with a site plan and photographs to illustrate the spatial configuration of an integrated settlement in relation to areas of the host community. We find that the South Sudanese refugees adjusted Bidi Bidi's spatial layout to better meet their needs. We also note that the loss of forested areas on and near the site has resulted in extreme hardship and contributed to gender-based violence. The third purpose of this article is to analyze and critique the land-based component of the self-reliance approach and note the importance of increasing integration between the refugee and host community populations.

## 2. Literature

In this section, we begin by clarifying the definitions of displaced persons, asylum seekers, and refugees. We draw from the extensive literature in anthropology, sociology, and political science to provide an overview of how refugee camps and integrated settlements vary in the freedoms and opportunities given to refugees. Finally, we highlight a study that compared indicators of well-being between refugees in an integrated settlement in Nakivale, Uganda with refugees in a "traditional" refugee camp in nearby Kakuma, Kenya to better understand self-reliance in this context.

### 2.1. Defining Displaced Persons

UNHCR differentiates among individuals who are internally displaced, displaced, seeking asylum or have obtained refugee status. An internally displaced person (IDP) is someone who has been forced to leave his or her home but still remains within his/her own country's legal borders. IDPs are not protected by international law because they remain under the protection of their own government. An asylum seeker is a person that has been forced to flee their home country due to fear of persecution (see United Nations 1951 Convention, amended in 1967 for the full definition). Once an asylum seeker has been registered, they obtain refugee status, which entitles the individual to international and host country rights (e.g., protection, aid, etc.) [1].

While the focus of this article is on integrated settlements and "traditional" refugee camps, it is important to note that millions of displaced persons do not live in either circumstance but live in spontaneous settlements or foreign cities. Akin to informal settlements, spontaneous settlements are regularly located in areas that are vacant because they are prone to natural disasters, such as flooding and landslides [18–20]. Displaced persons living in these spontaneous settlements may be without the legal right of occupation and have little to no access to emergency aid, physical infrastructure, or social services. Additionally, a significant number of displaced persons make their way to cities in foreign countries.

Although countries vary in their policies, generally in African nations, these urbanized displaced persons are considered illegal residents and are without access to emergency aid offered in camps.

*2.2. Refugee Camps*

Approximately 6.6 million people currently live in refugee camps [21]. The UNHCR defines refugee camps as "temporary facilities built to provide immediate protection and assistance to people who have been forced to flee due to conflict, violence or persecution. While camps are not intended to provide permanent solutions, they offer a safe haven for refugees where they receive medical treatment, food, shelter, and other basic services during emergencies" [22]. As noted above, in most refugee camps, refugees are not permitted to freely leave and return to the camp, and refugees cannot seek legal employment [8]. "Camps" are broadly associated with aid-dependent residents (those seeking food, water, shelter, etc.) who are confined in dense areas and heavily segregated from host populations (see Figure 1) [8,23]. In addition to being an efficient and effective method by which to deliver aid in emergencies, refugee camps also accomplish political goals such as containment and control [23]. Since many hosting nations consider displaced persons as an economic burden [24] that drains the limited services and resources away from long-term residents, the separated camps intentionally keep refugees from integrating with host communities [25]. While in policy many humanitarian organizations promote bottom-up participatory approaches within the camps, they are designed and located to restrict political agency [26–31] [2]. Thus, integration and self-reliance are difficult goals to achieve when refugees in refugee camps are not allowed to freely move or formally work alongside or with host communities.

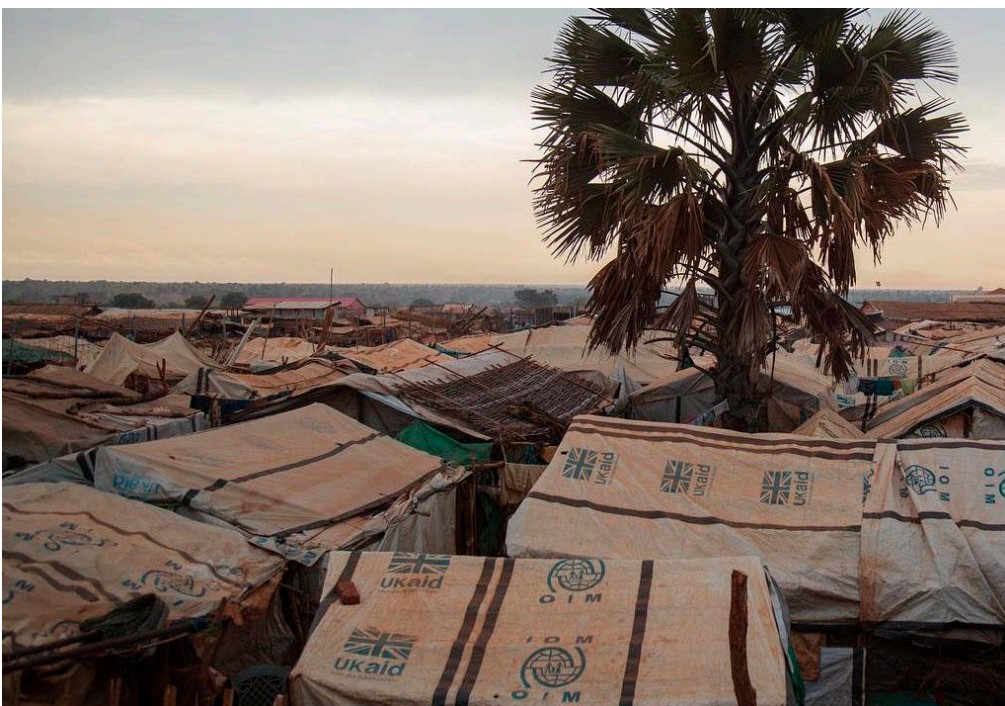

**Figure 1.** Traditional refugee camp design in Wau, Bahr el Ghazal, South Sudan Photograph taken by Bruno Feder, 2017.

In close coordination with UNHCR, willing host governments reactively determine the site location and layout of the refugee camp(s) or settlement(s) as violence in a neighboring country escalates. This is conducted with the guidance of UNHCR's emergency handbook and camp guidelines. Guidelines recommend locating refugee camps in areas that offer water access, provide stable soil conditions and shade, and have "waste management

capabilities". Generally, refugee camps are located no more than one day's walk from the border, so camps are close enough for displaced persons arriving on foot but are far enough to discourage militants from recruiting refugees to participate in the conflict. The camp location must also permit truck access for the delivery of supplies, with some refugee camps incorporating an airstrip.

Camps have a variety of public spaces, such as health centers, schools, vocational training centers, areas of worship, markets, roads, storage facilities, reception centers, and administration offices [32]. UNHCR has created basic minimum standards to guide the construction of infrastructure, community facilities, and individual plots [33,34]. For example, UNHCR recommends at least one health center with adequate staffing be available for every 20,000 people. In most cases, these guidelines are rarely achieved, as they are dependent on annual fundraising by UNHCR and other NGOs [9]. The first facility generally constructed in a new refugee camp or settlement is the registration center. Here, asylum seekers are registered as refugees with the host government and UNHCR. Once registered, they are recognized by international law providing them rights and entitlements, such as food and non-food items (e.g., a tarp for shelter, plot of land, etc.). It is worth noting that there are multiple critiques that question the way aid organizations implement programs in actual camp settings (see for example [28–31]).

### 2.3. Integrated Settlements and Self-Relance

Integrated refugee settlements have the potential to overcome some of the problems of refugee camps. In the literature, those who argue for integrated settlements instead of refugee camps offer two broad reasons. First, refugee integration can provide economic and social benefits that contribute to the host nation, such as increased GDP, agricultural production, and the increased provision of infrastructure funded by humanitarian agencies [9,10,35–39]. Second, unlike in camps, many refugees living in integrated settlements have the freedom to come and go at will and are legally permitted to work [8,9,13,14,35]. For example, refugees are able to access markets outside of the settlement for trade and income-generating activities that contribute to "self-reliance".

Globally, 85 percent of refugees are hosted in developing countries [1]. Furthermore, many refugees reside in highly impoverished host communities amidst residents who also struggle to survive. Therefore, the provision of aid, social services, and infrastructure for refugees can generate strong feelings of anger and resentment from residents of the host community toward the refugees. In an effort to recognize the unmet needs of host communities, UNHCR generally targets 30% of the humanitarian aid, programs, and infrastructure to local residents. For example, if new water boreholes are drilled, 30% of boreholes will be placed in the host community. Some elements of the development, such as roads and markets, generally benefit both communities.

Several studies have examined the impacts of refugees on host communities in African countries [10,36–38]. The majority of these assessments have focused on either the economic [10,36,38], educational [10,37], or health impacts [12,40] on host community members living near refugee camps. Most studies indicate that there are positive economic impacts on host communities due to the infusion of international funding that accompanies the influx of refugees. The creation of infrastructure, particularly roads and water, plus humanitarian aid, alters local trading, employment, and consumption patterns. However, some of the negative impacts associated with hosting refugees include increased competition over natural resources, such as water and trees (for construction and energy) [41], and employment (if refugees have the right to work). Other negative impacts include claims that refugees increase crime and petty theft [42] and introduce new illnesses. In sum, Jacobsen [39] argues that refugee-hosting governments must balance the direct and indirect political, social, and economical benefits that refugees provide while minimizing security threats and capacity issues. Host states also need to ensure adequate resources are provided where refugees are located (rural border areas) and ensure adequate staffing to coordinate services and provide security.

The "integrated settlement" concept has gained international attention and inspired the framework for UNHCR's Transitional Solutions Initiative (TSI) [43]. With the support of the World Bank and UNDP (United Nations Development Programs), the TSI is intended to integrate the provision of emergency services with the improvement of the host community and permit displaced persons to move toward self-reliance. Later in 2016, Uganda state officials committed to the Comprehensive Refugee Response Framework (CRRF) and pledged (1) to continue its integrated settlement approach, (2) to provide access to education and formal employment to refugees, and (3) to empower host and refugee communities [44]. This was later reaffirmed during the Kampala Declaration in 2017, which promoted the refugee self-reliance model. As defined by Harrell-Bond, integration is "a situation in which host and refugee communities are able to co-exist, sharing the same resources both economic and social with no greater mutual conflict than that which exists within the host community" [45]. "Self-reliance" in this context is defined as the "social and economic ability of an individual, a household or a community to meet essential needs (including protection, food, water, shelter, personal safety, health, and education) in a sustainable manner and with dignity. Self-reliance, as a programme approach, refers to developing and strengthening livelihoods of persons of concern, and reducing their vulnerability and long-term reliance on humanitarian/external assistance" [46]. Implicit in the notion of self-reliance in Uganda's integrated settlements is the belief that the majority of refugees will achieve self-reliance through agriculture on designated plots of land. It is also important to note here that this definition of self-reliance is that of aid agencies and romanticizes that notion of subsistence farming to meet the needs of refugee communities [47] (p. 36).

Betts and colleagues (2019) [10] have conducted some of the most thorough research comparing the indicators of the effectiveness of the Ugandan self-reliance approach with the traditional refugee camp approach. The researchers assessed the impact of the refugees' right to work and move, and the provision of land and access to markets on income levels, employment levels, mobility patterns, educational attainment, and the need for continuing assistance. The researchers surveyed over 8000 refugees in Uganda (where the integrated settlement policy exists) and Kenya (where the policy does not).

Most relevant to this study were their findings related to the impact of providing refugees with land and whether agriculture eliminated the need for aid and permitted self-reliance through the sale of surplus food. Because land is a finite resource and refugees had continued to arrive at the integrated settlement at Nakivale, Uganda, the size of plots had decreased as had the availability of plots for new arrivals. Initially, refugees were given a shelter plot measuring 15 m by 20 m as well as a cultivation plot measuring 50 m by 50 m. Over time, continuing demand required decreasing agricultural plots to 20 m by 50 m. Understandably, refugees had greater success in moving toward self-reliance with the larger plots. Another change over time related to the provision of land for agriculture was the decrease in soil fertility due to constant cultivation. So, as the agricultural plots became smaller, they also became less productive.

### 2.4. Context

In this section, we provide an overview of Bidi Bidi's context within the Yumbe District of northwest Uganda (630 km (391 miles) north of Kampala). The Yumbe District lies directly adjacent to the South Sudanese border. The southernmost counties of South Sudan are Kajo Keji and Moyo. Yumbe's topography is relatively flat with some rolling hills. The average annual rainfall is 1250 mm (50 inches), but it is concentrated into two periods. Agriculture occupies approximately 80% of the district [48]. Common crops are millet, potatoes, beans, and cassava. The remaining portion of the Yumbe District is forested. The forested areas includes acacia, cumbrietta, and fig trees [48].

According to Uganda's 2014 census, prior to the influx, 484,822 people lived in the Yumbe District and only 6% of the population lived in urban settlements [49]. The Aringa people have traditionally lived in this area, and the majority are followers of Islam. The Yumbe District has historically been one of the poorest regions in Uganda. The development

of infrastructure, schools, and economic diversification have lagged behind the country's averages. Most residents of Yumbe live in mud huts (tukels) with thatched roofs (Figure 2).

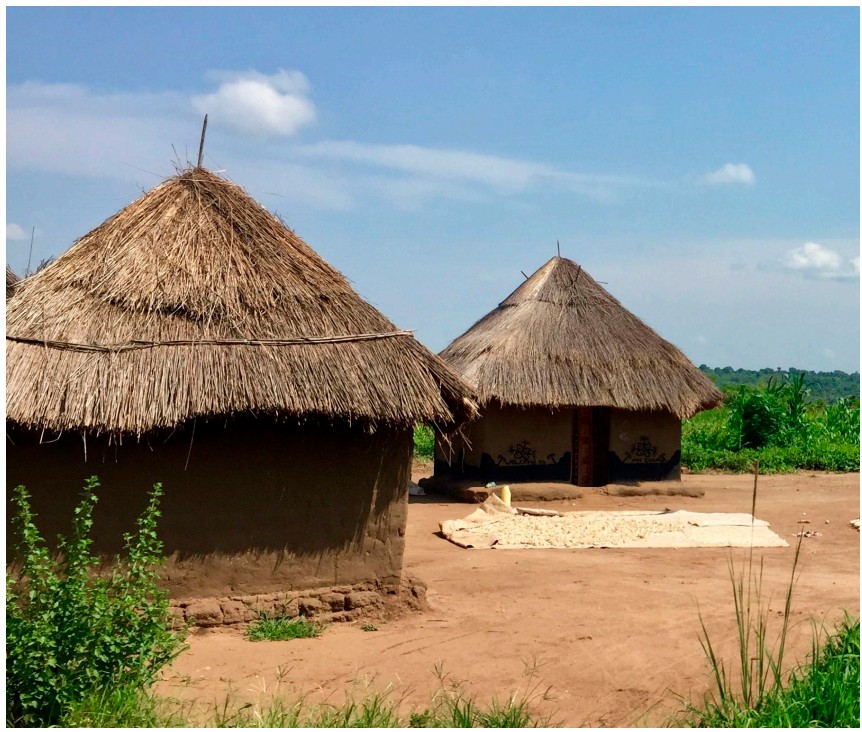

**Figure 2.** These are typical homes in the Yumbe District of northern Uganda. The residents are drying sorghum. Photograph taken by Tim Berke, 2019.

## 3. Methods

In summer 2019, the first author spent time during June through August accompanying staff of the non-governmental organization, Sustainable Children Aid (SCA), who introduced him to community members, local leaders, and government officials at the district and federal level [3]. Using a snowball sample method after initial introductions, the first author was able to formally interview 13 respondents, 9 men and 4 women (with IRB approval and the special permission of the Ugandan Prime Minister's Office). The first author interviewed 3 local Yumbe officials, 1 member of the Office of the Prime Minister, 3 traditional local leaders, 4 employees of different international aid organizations, one UN site planner, and one UNICEF protection officer [4].

All respondents were asked where they would like to be interviewed. Some interviews took place in respondents' offices, while others occurred in informal settings. For example, local district officials provided several "ride-alongs" in which the first author participated in day-to-day and special activities, such as tree planting. Additionally, local leaders were interviewed outside their homes. Recorded semi-formal interviews were conducted in English and lasted between 1 and 2 h [5].

The interviews were semi-structured and included questions about how refugees received land, coordination between and among service providers, lessons learned from establishing settlements for displaced peoples in emergencies, and many others.

The first author also attended and observed service provider coordination meetings, assessments, and public forums (e.g., World Refugee Day in Bidi Bidi). To validate and compare observation and interview data, we reviewed United Nations (UN) policy documents and reports, such as international standardized refugee camp planning guidelines, toolkits, and public reports about the Bidi Bidi settlement.

After transcribing the in-person interviews, the first author used an abductive approach of analysis by openly coding the transcriptions [50]. The initial phase included

examining surprising findings and dilemmas to generate themes. Once several themes began to emerge, (e.g., environmental derogation, issues of subsistence farming, coordination, and physical layout of the camp design), the first author used them to focus code the transcriptions while simultaneously reading the literature and relevant policy documents. Based on these sources of evidence, we began to understand how the planning and construction of Bidi Bidi had occurred, how the loss of trees negatively impacted refugees, why local leaders gave the refugees use rights to the land, and the strengths and weaknesses of the Ugandan integrated settlement policy.

## 4. Results

The results are divided into three sections. In the first section, we describe the establishment of the integrated settlement and the current conditions. In the second section, we summarize how the size of the allocated agricultural plots is decreasing and thereby threatening the viability of agriculture to meet refugees' subsistence needs and promote self-reliance. We also note that the rapid settlement process neglected to involve local leaders and district government officials. In our final section, we recount how site preparation practices removing trees exacerbated the loss of wood needed to provide refugees with materials for construction and fuel energy. The lack of wood has increased problems with gender-based violence and the lack of wood combined with the lack of sufficient land for farming has caused some refugees to return to South Sudan despite continuing violence.

### 4.1. Establishment of Bidi Bidi

On 8 July 2016, civil conflict broke out in South Sudan's capital between troops loyal to the South Sudanese President Salva Kiir and those loyal to Vice President Riak Machar. Subsequently, the violence displaced over four million South Sudanese. Roughly 1.9 million people were internally displaced while another 2.2 million South Sudanese people fled the country [51]. As the conflict spread southward in South Sudan, many displaced people walked to Uganda seeking safety and asylum.

After the conflict in South Sudan erupted in July 2016, Uganda state officials and UNHCR anticipated an influx of refugees into Northern Uganda. In response, the Office of the Prime Minister of Uganda (OPM) met with local authorities from the Yumbe district office to lobby for land to settle the newly arriving refugees. During this time, humanitarian actors (e.g., UN, NGOs, etc.) also began to coordinate emergency service delivery (the first author participated in four of these multi-sectoral meetings). In northern Uganda, most of the land is considered communal and governed by customary laws of a specific clan that has a historical claim to the territory. There is a "land chief" (or "landlord") that serves as the community representative overseeing how the land is used.

When asked why the local people would give land for the settlement, the Yumbe District official responded that the land was provided to help the refugees, with the expectation that development accompanying the refugees would also benefit the host community.

> "The land belongs to the people. It is not government land. And when the refugees came, the people willingly gave their land, freely without any money— because of the plight of the Sudanese refugees. When the refugees will go back to their country this land falls back to the landlords who gave it . . . [the local people benefit by getting access to] education, health, and then there are also other social services. Like they are giving water drilled, boreholes...They [the residents of the host community] get cheap food from the refugees, they buy. They can even exchange animals because communities have animals and the refugees can give them food [in exchange]. [The residents of the host community get] services like road construction, maintenance of the road and so on."

Additionally, this was supported by interviews with local landlords who suggested that local (at the district level) Ugandans could gain employment through international and national aid organizations. This was consistent with the findings of Vogelsang [52], who interviewed Ugandan landlords about their expectations of hosting refugees.

An official from the Office of the Prime Minister (OPM) recounted how they planned and constructed the settlement. Within days of the first influx of refugees, a UNHCR site planner created a basic master plan in "3–5 days". In the Director's description, he detailed the chaotic situation.

> "We (OPM) started demarcating that area and we produced, I think, over 2000 plots within one or two days. And then the refugees came... we spotted them along the road. And the UNHCR bulldozer started creating the roads inside, because it was a thicket. And besides that, there were streams. There were no bridges or culverts to pass on. And then, there were rocky areas but people had to be put on the land because the next day another 4000 people would come to the reception center . . . . So that's how we started it. And then later on, of course, the site planner (UNHCR) kept on drawing his plan but the movements of people were so great that within 4 months Bidi Bidi was already full with over 200,000 persons . . . we made 40 kilometers of roads inside there and . . . in the host community where the roads would pass. And then the rest of the infrastructure was just put there later on, the schools, health centers . . . "

Though the master plan provided guidance on general development (e.g., where roads, bridges, and residential areas would be located), it did not carefully consider the existing natural environment, and the recommended Environmental Impact Assessment was not performed. A preliminary report conducted by UNDP noted that the physical planning of Bidi Bidi had been limited due to financial constraints [53]. A later UNDP report noted concerns that the OPM and UNHCR had not fully involved the local district administrators/town councils in the land use planning and policies for the refugee settlement [53]. Local officials confirmed that they were not involved in the physical planning process and had not met the UNHCR site planners. While local landlords never mentioned frustration with refugees, they complained that OPM used the land without further consultation. *"Once the land is given the host community has no say over how it is used"*—Local Landlord [6].

District officials regularly expressed concerns about the longevity of the infrastructure and the lack of coordination in establishing where to locate facilities. Bidi Bidi, currently under the control of the OPM, recognized district administration as "token" parallel service providers who were eventually expected to assume full responsibility. The district officials suggested this temporary infrastructure, and district officials' lack of meaningful involvement could undermine the benefits of host communities.

> " . . . we have an emergency situation so most infrastructure put in is temporary. We need to put facilities that can accommodate people for the meantime. . . . from now onwards investments we need to do when in regards to infrastructure we need to put in permanent structures that can outlive the situation we are in in case the registration of refugees ends infrastructure needs to benefit the local community, as far as infrastructure development goes."

The national officials were aware of the concerns from district officials. When the regional director of the OPM was asked about permanence of the infrastructure he said, *"That is the proposal but you know these things are determined by money . . . The emergency started with temporary structures, so if money comes from donors we move to permanent. That is what has been going on."*

The Yumbe district officials wanted a detailed physical development plan to accompany UNHCR's rough master plan. However, the scale and speed of the emergency overwhelmed their efforts as well as their abilities to coordinate the hundreds of various stakeholders (e.g., NGOs, UN agencies, OPM, host community, refugees, etc.). District officials believed that a context-specific physical development plan would have helped them guide and coordinate the numerous NGOs developing structures (e.g., schools, health clinics, etc.). Waters [28] discusses the challenges of one-size-fits-all planning and the frustration local administrators often face when implementing them. District officials also

stated that if local officials and local leaders had been consulted by the disaster management committee, a better plan would have resulted. For example, the Yumbe district planner said:

"One key recommendation should have been administration whereby you want to start NGO, camp, IDP camp, one thing you should ensure is such area, should have physical development, structured plan."

"The minister of disaster came out clearly and said we need to get areas where we can relocate people, and how do we relocate these people? You need to plan those areas. Meaning there is some bit of disaster management plan, but whatever we do in Uganda here has been in a piecemeal way. We have not been having something comprehensive. We just come and do something in a piecemeal way, to arrest a situation. And when this situation is done, we tend to forget, and not until we also succumb to a similar situation . . . But I think, what has happened with Bidi Bidi has taught all of us, has given us a lesson and we're now looking at handling issues in a comprehensive way."

"As a physical planner, the physical development plan is very key because it is going to streamline all the other small activities going to take place there. It will lay out the infrastructure plan . . . to be very clear."

While district officials expressed concerns about the longevity of some of the infrastructure, they were encouraged by the amount of development in Bidi Bidi (see Figures 3–5). When a government official was asked about the future of Bidi Bidi, he suggested that if permanent structures were implemented and maintained, Bidi Bidi could turn into a model city.

"Actually, it has beaten the town because of the beautiful structures which they have put [in]... the already existing roads, eh, should be well maintained. If they get the funds to allow they can even tarmac the roads . . . And then if we have proper physical planning, that... can involve the local government in planning and so on, the settlement can become a model city in the future."

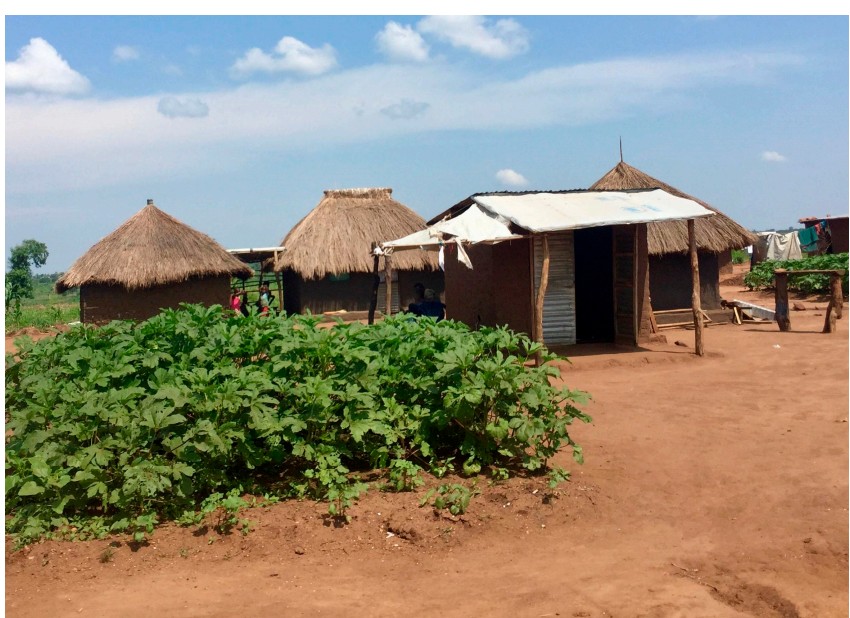

**Figure 3.** These refugee homes are indistinguishable from the homes of the host community's residents. In addition to farming their allocated plot, this family has established a small store in front of their homes. Photograph taken by Tim Berke, 2019.

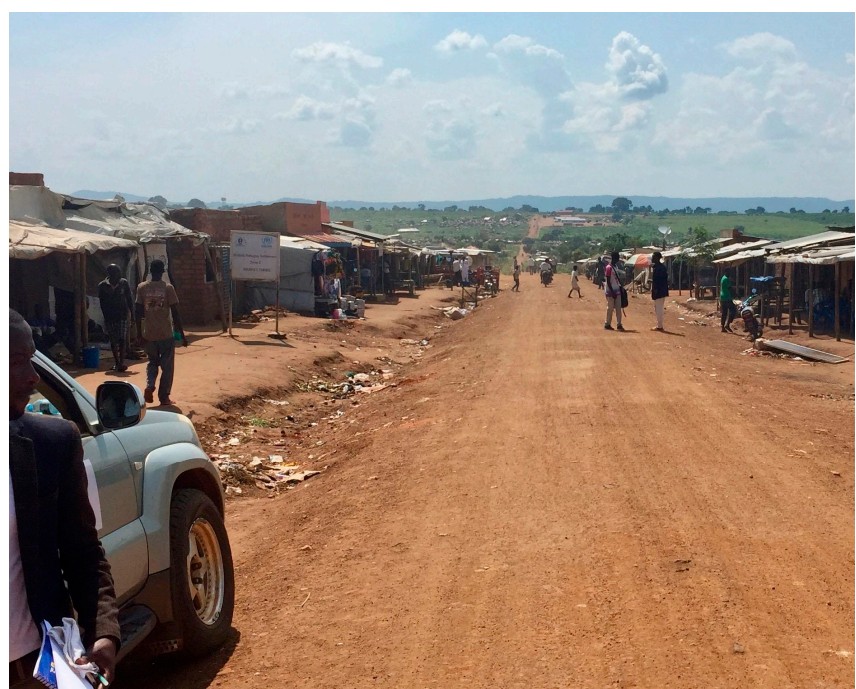

**Figure 4.** The market in Zone 2 is the largest market in Bidi Bidi. On this day, a national holiday, the market is quiet. Photograph taken by Tim Berke, 2019.

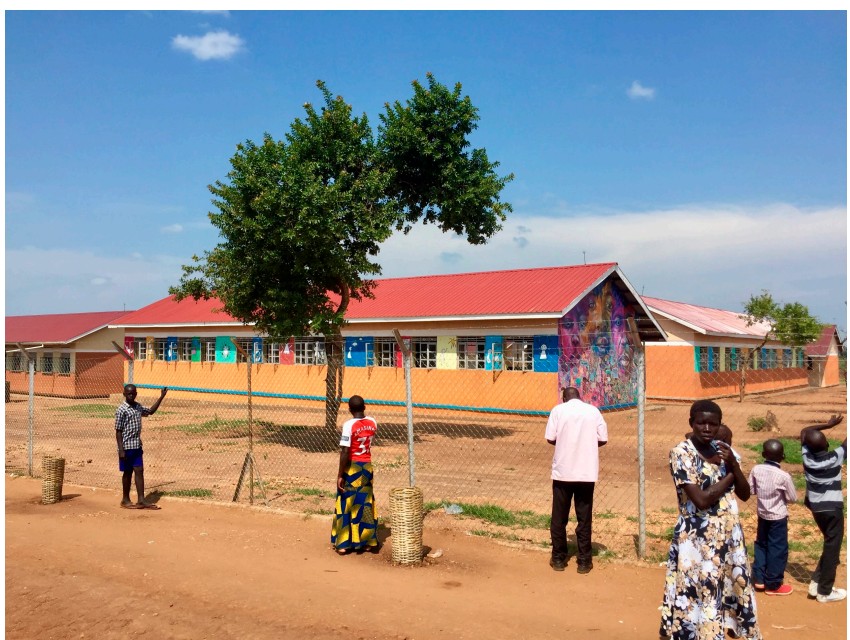

**Figure 5.** This is one of the elementary schools in Bidi Bidi. Photograph taken by Tim Berke, 2019.

Another official suggested that Bidi Bidi could become the Central Business District (CBD) of Yumbe, "Well with the trend of things are going there I would say I could see it becoming a very big city. Very busy town. In the situation of Bidi Bidi [it is] even having what kind of conversion to [Central Business District] CBD". He then provided many examples of businesses that were growing within the camp (Figure 4) such as transportation hubs.

The district OPM official explained that they had incorporated lessons learned from past settlements, such as ensuring that there are cemeteries. "In the past we were not

planning for cemeteries but this time we had to plan for cemeteries because it was causing a lot of stress in other settlements."

### 4.2. Current Conditions

Refugees living in Bidi Bidi now constitute 1/3 of Yumbe's population. Bidi Bidi now occupies approximately 32 percent (764 km$^2$ (295 mi$^2$) of 2393 km$^2$) of the Yumbe district and is divided into five zones (Figure 6). According to UNDP, 83% of the settled areas of Bidi Bidi are now fully occupied [53] (p. 7). The camp is sectioned into five different "zones". Zone 1 is located in Romogi (sub-county), Zone 2 in Kochi, Zone 3 in Kululu, Zone 4 in Odravu, and Zone 5 in Ariwa. Each zone has areas for residential and agricultural uses, a small market area, health facility, schools, and 10 to 20 "villages". The villages could be likened to neighborhoods. In between the five zones live residents of the host community, and the Figure 6 map illustrates the smaller areas of host residents relative to the areas for refugees. The market in Zone 2 has evolved into the largest of the area's markets. Based on our efforts to geolocate UNHCR maps, we estimate that the refugee settlement area measures 55 km$^2$ (7%), and the designated agricultural plots constitute 49 km$^2$ (6%). Large portions of the area remain open as traditional hunting grounds.

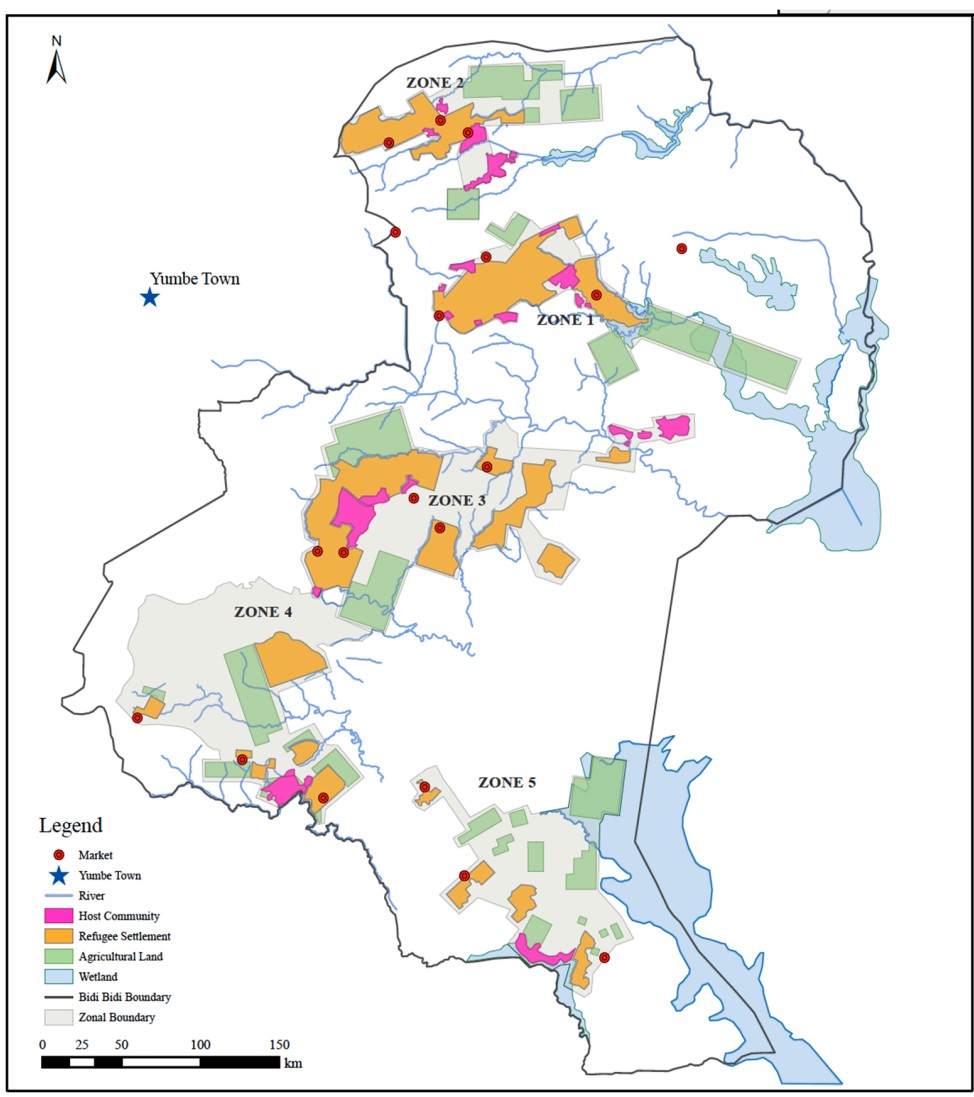

**Figure 6.** Map of the Bidi Bidi settlement. The map was created using UNHCR documents and materials dated 16 May 2019.

When planning refugee settlements, UNHCR applies a standardized grid pattern for residential space. Therefore, in Bidi Bidi, a grid pattern was used to organize the agricultural plots and shelter sites. When first allocating plots of land in a grid layout, UNHCR and OPM assigned shelter sites for the house adjacent to the 50 m by 50 m agricultural plot. The intention, according to a UNHCR site planner, was that people could *"eat from day one, have that ability to be self-sufficient, generate an income"*. However, as land was assigned and the influx of refugees continued, OPM reduced this plot size to 30 m by 30 m to accommodate new arrivals.

Generally, refugee households now receive a 30 m by 30 m plot [53] (p. 4). Using United Nations Food and Agricultural Organization (FAO) data, UNDP conducted an analysis to assess the potential productivity of a 30 m by 30 m plot, assuming fertile soil and maximum yields. In the analysis, they found that plots of this size could not meet the dietary needs of refugees in the absence of food rations [53] (p. 9). The same report suggested that the quality of the land given to refugees varied widely, with some rocky and infertile areas amidst fertile areas. These findings challenge the assumption that refugee households could become self-reliant through subsistence farming [53] (p. 7). This was reinforced by a recent UNHCR report (April 2020) stating 97% of Bidi Bidi households were still receiving food assistance [5].

In addition to insufficient fertile land, weather variation complicates refugees' efforts. Uganda has two rainy seasons, usually from September through December and February through April. The dry seasons (when harvesting takes place) are generally in January and July. In 2018, Uganda experienced a drought that resulted in extensive crop loss and constrained access to water, which seriously impacted refugees [54]. Climate change is expected to exacerbate drought and flooding and either weather condition could significantly reduce refugees' ability to rely on subsistence agriculture.

Since the camp's establishment in 2016, the refugees have modified its layout to better meet their needs [7]. A UNHCR site planner noted that many people did not live on the allocated settlement site adjacent to their agricultural plot but opted to cluster their homes near friends and relatives and walk to their fields.

> "The settlements are really sprawling out . . . People while initially may have been allocated these orthogonal [plots], rather generous in terms of size, but people are actually more densely . . . clustered around trading center hubs. And they would leave for . . . agriculture, they separated entirely, and they'd farm different parts of the landscape. So...what I'm trying to say is that people [are] not really sticking to the expected grid . . . the tendency was to have the thriving more urban center, the residential and trading and markets, etc. And then to have buffers between these hubs, you actually [have more] agricultural land . . . It wasn't planned. But that has been the tendencies, so people would be allocated these parts, but then they would actually not build on them. They would use those parts purely for agriculture. And then they would build next to neighbors and friends and extended relatives, families who'd already crossed circles".

### 4.3. Deforestation

While we focused our interview questions on how refugees were using land and whether it was sufficient to support self-reliance, we quickly learned that the lack of another natural resource, trees, was imposing considerable hardship on both the refugees and host community residents. In addition to using trees for construction materials (for homes and homesteads), wood is relied upon for cooking [8]. As mentioned above, large areas of trees were removed in the initial construction process to "clear" the settlement area. Interviewees noted that greater care should have been taken to preserve some of the stands of trees on the site during the initial construction phase. Post-construction need by refugees and host community residents had quickly depleted surrounding wooded areas [9].

Generally, South Sudanese women and girls are responsible for collecting firewood to cook. One of the main concerns when collecting firewood is the potential threat of gender-

based violence, specifically rape. When the Bidi Bidi settlement was first established in 2016, refugee women and girls could walk a few minutes to safely collect firewood. According to the Yumbe District Staff member, this changed rapidly. The following year (2017), women and girls had to walk 5 km and then in 2018, ten kilometers. As the settlement grew and the demand for firewood increased, the need to walk farther and farther created a situation in which women and girls were exposed to longer, unsafe daily journeys.

In interviews with two social workers and a protection manager employed by a national NGO, they emphasized that deforestation was leading to gender-based violence, death, and relocation of refugees. One of the social workers explained *"If you go to the bush to collect firewood—the people from the host community...they attack you and they kill you. So, they [the refugees] also fear that."*

From the interviews, we learned that a small number of refugees were returning to South Sudan despite danger from the continuing conflict. When asked why people would risk going back (to the IDP camps in Kajo Keji), the national NGO employee responded that these refugees felt they had better chances to collect firewood, and the land was more fertile to farm, which meant they could generate an income by selling produce (and not buying fuelwood). For some refugees, having a higher degree of self-sufficiency in the conflict zone outweighed the offerings of the integrated settlement outside the conflict zone.

## 5. Discussion

While there is literature on Uganda's refugee policy and self-reliance [6,9–16], it focuses on the policy and not the planning process of establishing the physical settlement and the resulting outcomes. Additionally, while there have been some UN evaluations of Bidi Bidi, few academic investigations have been published despite its position as the largest refugee settlement in Uganda (see [56]).

This research has documented some of the changes that have occurred in Uganda's Yumbe District since the establishment of the Bidi Bidi integrated settlement in 2016. The district population has rapidly increased from 484,822 to approximately 715,887 in four years [53]. The local leaders have given use rights to a portion of their land in exchange for the investment, infrastructure, local employment, and social services that they expect to accompany humanitarian assistance. The planning, plot survey, and initial road construction occurred in approximately 3 days. This rapid settlement process neglected the involvement of local tribal leaders and district government officials and has resulted in environmental degradation. The current settlement is approximately 765 km$^2$ (295 miles$^2$) and it is divided into five zones/sub-areas. While the total area of Bidi Bidi is relatively large, the large numbers of refugees and the diversity of their skills have meant that the land-based approach to self-reliance is insufficient.

We conclude by extracting three key findings from our research and suggest policy and practice modifications to better serve refugees and residents of the host community in a more environmentally sustainable way. These findings concern expanding the paths to self-reliance, improving "integration" for both refugees and residents of the host community, and protecting the area's environmental resources to permit their continued use.

### 5.1. Plot Allocation and Reality of Self-Reliance

The first conclusion concerns the questionable reality of the self-reliance program based on agricultural plots due to limited land availability and the diverse backgrounds and abilities of refugees. Hunter [47] suggests the UNHCR self-reliance strategies place a romantic and unrealistic emphasis on subsistence farming to meet the needs of refugee communities [47] (p. 36). While we support Betts and colleagues' [10] finding that "a functioning land allocation system can be an effective means to support refugees from agricultural backgrounds", our research found that by subdividing the agricultural plots to meet demand, the smaller plots weren't sufficient to provide food for the family and generate a surplus that could be sold. Additionally, as refugees stay for longer periods, family sizes get larger, and soil fertility decreases [13]. Land is a finite resource of varying

fertility. As revealed by Werker [14], we found that the land provided to refugees was often not as fertile as host areas. Within refugee integrated settlements, the number of agricultural plots should be limited in number and focused on fertile areas. Reducing everyone's plot size gradually to meet the needs of new arrivals is removing this as a viable route to self-reliance. This highlights the need for diversifying the livelihood options beyond crop cultivation. For example, Grosrenaud et al. (2021) [57] found that if refugees are active participants in preserving the environment in the settlement agroforestry, it improves all livelihoods and increases nutrition rates.

Working with refugees and members of the host community in a collaborative planning process may generate an expanded array of options. Jahre et al. [32] conducted a comprehensive assessment of the integrated settlement approach in Kenya, Ethiopia, Greece, and Turkey. They found that the top-down approach existed during the initial planning and construction phase of the integrated settlement. However, as the settlement matured, more bottom-up participatory decision making began to occur. Werker [14] argues humanitarian agencies and planners can reduce the constraints refugees experience in accessing the market by either providing transportation or locating the camps closer to existing markets. Considering refugees' likelihood to experience an extended stay in the integrated settlement, workforce development that helps move residents toward self-reliance and fosters access to markets will lessen the amount of long-term support required.

### 5.2. The Need for Integration

Our second and third conclusions echo the words of a district staff member. When asked to share the lessons he had learned from his experiences in Bidi Bidi, he identified the need to improve social integration and protect the environment for the settlement's long-term viability. Our second conclusion addresses the issue of integration from three perspectives: (1) the lack of integration between the local planners/administrators and host community leaders with OPM, the UNHCR, and non-governmental organizations, (2) the lack of integration between the refugees and residents of the host community, and (3) the lack of integration of the refugees into Ugandan citizenship.

Our research found that local and district administrators/town councils and host community leaders were not involved in the on-going planning and management of Bidi Bidi. Interviews with individuals working in the OPM, UNHCR, non-governmental organizations, and local authorities all recognized that this omission resulted in many problems. These included the district officials and host community leaders' unmet expectations to provide input on land use planning and preservation of the wooded areas. In part, the initial disconnect may have occurred because of the speed of the construction. However, once some stability had been achieved, efforts should have engaged the local planners and people. Policy decisions intended to benefit local officials and host community members are being made in Kampala and Geneva. The issue of top-down planning is an enduring problem. If goals, such as self-sufficiency, continue to be set and defined by international agencies and donors (Waters, 2001) without an understanding of how refugees and host communities live, programs will continue to be ineffective [29]. The Regional Durable Solutions Secretariat (ReDSS), a consortium in East Africa that focuses on durable solutions policy development (e.g., service integration), suggested: "*There should be a greater investment of time and resources in settlement and site planning, including attention to building local capacities to participate more effectively in these processes*" [58].

Another form of integration concerns integrated service provision. Our findings suggest that district officials had concerns over the longevity of infrastructure and the lack of coordination around siting facilities for social services such as education and health care. Integrated service provision is based on the concept that refugees and residents of the host community will have access to the same quality of facilities and services. In Uganda's refugee policy, 30 percent of refugee services are intended to serve the host population, and in most refugee settings, humanitarian actors operate distinct parallel services for refugees and for residents of the host community. In addition to maintaining separation between

the refugees and residents, operating parallel services can be costly and inefficient [59]. Currently, OPM, with the support of UNHCR, directly oversees all refugee processes and service delivery. While research indicates that integrated services are resulting in some positive impacts for both host and refugees, the structure and administration of services delivery has not been decentralized to the district level. Similar to Tuepker and Chi [12], ReDSS [58], and Jacobsen (2002) [39], we recommend that as the settlement matures, more responsibilities for service delivery should fall upon the appropriate ministries, districts, and local administrations versus OPM and UNHCR. Expanding integrative services into administrative structures in Bidi Bidi would support the host community's buy-in, reduce any parallel service delivery, and promote community integration.

Finally, the truest form of integration would be offering refugees steps toward legal citizenship in Uganda. While a topic beyond the focus of this paper, Uganda's settlement policy does not provide a "durable solution" (citizenship) for protracted refugees, nor do they incorporate refugees' views of durable solutions [60,61].

*5.3. Protecting the Environment from Irreparable Damage*

Our third conclusion concerns the need to recognize that both refugees and residents of the host community rely on the environment beyond agricultural land. In the initial stages of establishing Bidi Bidi, greater care should have been taken to preserve trees. One of the biggest problems in Bidi Bidi is now the lack of wood for construction and fuel energy. Recent reports have estimated that each household would need a 50 m by 50 m plot to grow trees to meet their fuelwood needs [55]. However, plots have already been reduced to 30 m by 30 m and are intended for agricultural and residential purposes. The local officials stated that while the environment was neglected in the first two years of Bidi Bidi's establishment, it is now recognized as a top priority amongst all stakeholders (e.g., host, refugees, government, and humanitarians). Tree planting programs are currently underway.

The Food and Agriculture Organization of the United Nations (FAO) has created guidelines for safe access to firewood and alternative energy in humanitarian settings to reduce gender-based violence. The guidelines advocate for designated areas for woodlots, efficient stoves, and farming that produces both food and fuel [62]. Tree planting activities can be used to support peacekeeping activities and reduce tensions between refugees and residents. In an interview, a local official recommended including tree seedlings in the packet of non-perishable emergency aid supplies distributed to refugees upon entry.

This finding highlights the need to consciously recognize that the initial settlement is likely to host refugees for extended periods and will have lasting impacts on the host community. Therefore, the integrated settlement needs to balance short-time emergency relief provision with the land use and environmental planning necessary for a viable settlement. We recognize this is no easy task and it may require a phased approach where refugee families move to "permanent" locations within the settlement after their initial period of stabilization. With this phased approach, refugees could express their needs and preferences related to livelihood opportunities and living near family/former community members. Currently, refugees are assigned their permanent place upon registration, without their input.

Integrated settlements are not a simple solution to the refugee "problem", but they can offer refugees opportunities and empowerment while providing host communities with needed improvements and economic investment. While continuing research can identify effective modifications to the integrated settlement approach, this needs to be performed concurrently with global discussions of collective responsibility of all countries regardless of proximity. This discussion of collective responsibility was a major aspect of the 2016 New York Declaration [50]. All UN member nations agreed to acknowledge their global responsibility by accepting refugees into their own countries as well as to financially support other countries, such as Uganda, that provide many refugees protection.

**Author Contributions:** For this article, both T.B. and L.L. contributed to the conceptualization, formal analysis, methodology, and writing and editing of the original manuscript draft. T.B. was responsible for funding acquisition, investigation and data collection, and project administration. All authors have read and agreed to the published version of the manuscript.

**Funding:** International Institute, University of Michigan, provided funding (USD 1500) for this research. The funder had no involvement in the preparation of this manuscript.

**Institutional Review Board Statement:** The study was conducted in accordance with the Declaration of Helsinki, and approved by the Institutional Review Board (or Ethics Committee) of the University of Michigan (HUM00164056 on 20 May 2019).

**Informed Consent Statement:** Informed consent was obtained from all subjects involved in the study.

**Data Availability Statement:** Not applicable.

**Acknowledgments:** We would like to acknowledge and thank Meixin Yuan, student at the University of Michigan, for her work in creating Figure 6 and Charles Wani, Director of Sustainable Children Aid (SCA) for sharing his time and networks during data collection in Uganda.

**Conflicts of Interest:** The authors declare no conflict of interest. Additionally, the funders had no role in the design of the study; in the collection, analyses, or interpretation of data; in the writing of the manuscript, or in the decision to publish the results.

## Notes

1. This can be a challenging and complex process, but in countries such as Uganda that experience largescale influxes of displaced persons they receive refugee status on a prima facie basis. This means the state recognizes refugee status due to "*basis of the readily apparent, objective circumstances in the country of origin giving rise to exodus*" [17].
2. While the camps restrict political agency, they open new spaces for resistance, contestation, and new possibilities for political action [26,27].
3. The first author established social and professional networks working with SCA and other organizations on the border of South Sudan and Uganda for four years.
4. Although the first author interacted (e.g., lived and ate) with refugees, time constraints prevented acquiring approval from IRB and OPM to formally record and interview refugees. Although we incorporated participant observation, we acknowledged the limitations of not formally intervieing the refugees themselves. We are actively persuing permission to conduct such interviews.
5. Although formal interviews were in English, there were times in which Arabic (Juba) was used in interactions with refugees.
6. Based on many conversations with residents, resentment was not expressed toward refugees as they understood their plight. However, some of their resentment was directed toward the office of the prime minister for lack of adequate "compensation" for their land.
7. While most refugees in Bidi Bidi are South Sudanese, they do not share a single, homogenous culture. While some of the refugees come from urban areas such as Juba, the majority come from rural areas and are either agriculturalists or pastoralists and self-identify as farmers [53]. Even though many refugees were farmers, conflicts have arisen between "agriculturalists" and "pastoralists".
8. Recent assessment conducted by UNHCR estimated that on average 3.5 kg of fuelwood was used per person per day in Bidi Bidi [55].
9. To demonstrate the importance of deforestation being experienced within and outside of Bidi Bidi, World Refugee Day on 20 June 2018 was held under the theme of "*Take a step #withrefugees protect the environment*". Numerous local, national (including the Ugandan vice president), international and refugee leaders publicly discussed the pressing use of deforestation in Bidi Bidi. This was followed with a symbolic joint activity of planting 10,000 trees by both host and refugee community members (the first author took part in public discussion and activity).

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
