# Peer review of "Using Land to Promote Refugee Self-Reliance in Uganda"

_land, doi:10.3390/land11030410_

Round 1

Reviewer 1 Report

The article "Using Land to Promote Refugee Self-Reliance in Uganda" deals with a relevant topic and fits the journal’s aims and scope. The main concepts authors use are adequately defined. However, to avoid confusion, I would recommend using the term asylum seeker instead of asylum searcher (line 91). The article is logically structured and fulfills the stated purposes, or main goals (p. 2).

My main objection concerns the section on Methodology which is too basic and should include some more information on the context of the fieldwork, interview type, duration and language, gender of the respondents and, not least importantly, information on qualitative data analysis method. It would also be good to explain in more detail why the permission to interview refugees was not granted, as it would have been useful to include their perspective as well.

In addition, there are some minor points throughout the text that should be addressed.

The first sentence of the article (lines 26-27) ought to include technological disasters as one possible generator of involuntary migration.

The sentence on p 1 (lines 32-33) should be clarified - does it relate to the time spent in refugee camps, or "in asylum" generally?

Please check grammar in line 94.

The last paragraph in section 2.1 (96-105) should be further contextually qualified - e.g. in African countries, or in Uganda - because it is not universally applicable (e.g. European experience with hosting Syrian refugees varies from country to country, and urbanized displaced persons are not a priori considered illegal).

The footnote 2 has repeated text.

The sentence (170-171) should state "in African countries" as there are many more studies focusing on the impact of refugees in other geographic contexts.

The sentence (180-181) ought to include the author’s surname.

When first mentioned in the text, CRRF (line 191) should be written in full.

FAO should be referenced correctly (Food and Agriculture Organization, line 425).

Generally, the article offers a valuable insight into refugee integrated settlement in Uganda, and analysis of intended and unintended consequences of the land-based approach to achieve refugee self-reliance in a particular case.

Author Response

Thank you for your time and helpful feedback. Please see the attachment with our responses. 

Reviewer 2 Report

General Comment.  This paper describes an interesting process of how refugee camps/settlements were established in Uganda after 2016.  The paper does a good job of this.  As the article points out, Uganda has generous asylum policies, which have permitted the admission and resettlement of very large numbers of refugees.  It would be nice if other countries were this generous.

One ”weakness” as such is that this paper draws primarily on UNHCR and other official documents, and interviews with UNHCR and government officials.  The refugee viewpoint is not directly found here, albeit for reasons elaborated on in the paper, and should be elaborated on in the paper more thoroughly.

A second weakness as such, is that the perspective is primarily from the field of Geography and Planning.  But the bulk of the “refugee camp” literature is in anthropology, sociology, and sometimes political science.  You can’t cover all of these in this paper, but some acknowledgment would be nice.

A strength of the article is the photographs!  I appreciated those.

General Notes

1) Need to make careful distinction between settlement, and refugee camp in the first pargraphs.  Don’t just rely on UNHCR definitions, though of course these are relevant. 

2) Section 3—Too UNHCR-centric. There is much good about HCR, and much to criticize about their policies which tend to be UN-centric, rather than refugee-centric.  Anthropologists in particular have raised this critique for decades, and a few are mentioned below.

3) Lines 126-167. describe the UNHCR ideal of a refugee camp, from their handbook.  There are many critiques that question the actual on-the-ground ways this is implemented. You cite Turner’s article about refugee camps in the paper.  He also has an excellent critique in that in his earlier book about Lukole camp in Tanzania, Politics of Innocence (2010).  Also see Liisa Malkki’s book The Purity and Exile, about settlements in Tanzania.  Ban Vinai The Refugee Camp by Lynelyn Long is about Thai camps.  Mark Sommers Fear in Bongoland is about Burundians in western Tanzania, and Dar Es Salaam.  There is also plenty written about Syrians, Iraqis, etc., in Turkey and other countries in the Middle East. 

4) For a more general critique of UN agencies, see Severine Autessere, Peaceland (2014).  Also you already cite Simon Turner’s article (2014) on refugee camps.  This should be more prominent.

5) Line 170+ Plenty of articles on how host communities are effected by refugee crises.  You list a number.

6) Methods Section.  How do you make up for the refusal of permission to interview refugees htemselves? Did this introduce a potential bias to your findings?  Why did the Prime Minister’s office refuse permission?  Such refusals are unfortunately normal in refugee studies.

7) Around Line 346.  The planning for such a large settlement was done very quickly! This is often done in emergency circumstances, with mistakes often becoming apparent only months or years later.  UNHCR has been critiqued around the world for this type of planning.  To UNHCR’s credit they have developed “camps in a box” or “school in a box” type of infrastructure.  But this too creates situations that are not necessarily conducive to good use of available land, etc.  See Tony Waters, Bureaucratizing the Good Samaritan 2001.  Lines 441-454 acknowledge the limitations of this one-size-fits-all planning.

8) The point Hunter makes (line 515) about the romanticization of subsistence agriculture is an important one.  This is done by both refugees and UNHCR all over the world.  This point needs to be highlighted earlier, and more forcefully.

9) Lines 552-565.  This complaint about top-down planning is an enduring one.  I have heard it since becoming involved with such planning in 1982.  UNHCR, governments, NGOs, etc., have always had a tough time listening to refugees.  See Bureaucratizing the Good Samaritan, and Peaceland, both cited above.

Author Response

Thank you for your time and helpful feedback. Please see our responses attached. 

Reviewer 3 Report

Both the introduction and the context section explain in detail the reality from which this study is based, as well as the objectives of the study. A review is made of the problems of the geographical region where this refugee camp is located and the process of its construction is explained, as well as the role of the actors involved, international institutions, NGOs, local administration, ...

I find the research design very interesting, based on interviews with different actors, thus gathering the opinion of all those involved, which helps to better understand the problems and allows the authors to propose solutions based on reality.

The results are presented according to the objectives set out in the introduction, which facilitates the understanding of the work.

Clearly, the conclusions are based on both the interviews and the literature review. The bibliography is very up to date and contains a large number of citations.     

I would mainly like to emphasise that I found the topic and the approach of the research very interesting and topical. I believe that work like this can help to improve the problems and realities of today's difficult situation.

Author Response

Thank you for taking the time to review our manuscript and for the encouraging feedback.